# The Evaluation of Glioblastoma Cell Dissociation and Its Influence on Its Behavior

**DOI:** 10.3390/ijms20184630

**Published:** 2019-09-18

**Authors:** Veronika Skarkova, Marketa Krupova, Barbora Vitovcova, Adam Skarka, Petra Kasparova, Petr Krupa, Vera Kralova, Emil Rudolf

**Affiliations:** 1Department of Medical Biology and Genetics, Faculty of Medicine, Charles University, Simkova 870, CZ-500 38 Hradec Kralove, Czech Republic; vitovcob@lfhk.cuni.cz (B.V.); kralovav@lfhk.cuni.cz (V.K.); rudolf@lfhk.cuni.cz (E.R.); 2The Fingerland Department of Pathology, Faculty of Medicine and University Hospital in Hradec Kralove, Charles University, Sokolska 581, CZ-500 05 Hradec Kralove, Czech Republic; marketa.valkova@fnhk.cz (M.K.); petra.kasparova@fnhk.cz (P.K.); 3Department of Chemistry, Faculty of Sciences, University of Hradec Kralove, Hradecka 1285, CZ-500 03 Hradec Kralove, Czech Republic; adam.skarka@uhk.cz; 4Department of Neurosurgery, Faculty of Medicine and University Hospital in Hradec Kralove, Charles University, Sokolska 581, CZ-500 05 Hradec Kralove, Czech Republic; petr.krupa@fnhk.cz; 5Institute of Experimental Medicine, Czech Academy of Sciences, Videnska 1083, CZ-142 20 Prague 4, Czech Republic

**Keywords:** glioblastoma multiforme, cell isolation, temozolomide, resistance

## Abstract

Purpose: Primary cell lines are a valuable tool for evaluation of tumor behavior or sensitivity to anticancer treatment and appropriate dissociation of cells could preserve genomic profile of the original tissue. The main aim of our study was to compare the influence of two methods of glioblastoma multiforme (GBM) cell derivation (mechanic—MD; enzymatic—ED) on basic biological properties of thus derived cells and correlate them to the ones obtained from stabilized GBM cell line A-172. Methods: Cell proliferation and migration (xCELLigence Real-Time Cell Analysis), expression of microRNAs and protein markers (RT-PCR and Western blotting), morphology (phase contrast and fluorescent microscopy), and accumulation of temozolomide (TMZ) and its metabolite 5-aminoimidazole-4-carboxamide (AIC) inside the cells (LC-MS analysis) were carried out in five different samples of GBM (GBM1, GBM2, GBM32, GBM33, GBM34), with each of them processed by MD and ED types of isolations. The same analyses were done in the A-172 cell line too. Results: Primary GBM cells obtained by ED or MD approaches significantly differ in biological behavior and properties of these cells. Unlike in primary MD GBM cells, higher proliferation, as well as migration, was observed in primary ED GBM cells, which were also associated with the acquired mesenchymal phenotype and higher sensitivity to TMZ. Finally, the same analyses of stabilized GBM cell line A-172 revealed several important differences in measured parameters. Conclusions: GBM cells obtained by MD and ED dissociation show considerable heterogeneity, but based on our results, MD approach should be the preferred method of primary GBM cell isolation

## 1. Introduction

Glioblastoma multiforme (GBM) represents the most aggressive form of glial tumors, characterized by extensive genetic, as well as epigenetic, alterations leading to multiple changes in biological behavior of malignant cells [1,2]. Complex and aggressive phenotypes of GBM cells, as well as their localization, greatly limits our intervention strategies. Current standard GBM treatment thus includes surgical resection, followed by radiation, and co-administration of oral chemotherapy agent, temozolomide (TMZ) [3]. Unfortunately, different studies reported that brain concentrations of TMZ reach only 17–20% of those in the blood [4,5] and the possibility of their further increase is limited due to TMZ-related toxicity effects, including leukopenia or thrombocytopenia, with a possible increased risk of opportunistic infections [6]. Moreover, GBM cells also show chemoresistance to TMZ, which is often present shortly after the initiation of the treatment. This chemoresistance is likely multifactorial, including changes in the cell cycle, upregulation of mismatch repair genes, and additional resistance mechanisms involving resistance’s markers expression [7]. Besides the above-mentioned limitations of the standard care strategy of GBM, there are additional therapy-related concerns, which include for instance the diffuse infiltrative nature of the tumor based, at least in part, on invasiveness of GBM cells. These cells typically invade up to several centimeters away from main tumor mass and can even cross into the contralateral hemisphere. GBM also contains a subset of glioma stem cells (GSCs), which may have an increased invasive potential and are thought to be more resistant to radiotherapy and chemotherapy compared to non-stem tumor cells. This hypothesis is corroborated by the observations that recurrent and chemotherapy-resistant tumors are enriched in the highly invasive mesenchymal GBM subtype. Gliomas seem to prefer single-cell migration and invade over longer distances than other tumors that metastasize in the brain [8].

Since current therapeutic options of GBM are rather limited and, consequently, the prognosis in most cases dismal, new treatment modalities are urgently needed. Primary cell lines derived from glioblastoma specimens are a valuable tool for evaluation of tumor behavior or sensitivity to anticancer treatment. Recent evidence, however, indicates that classically established cell lines from different tumors, including GBM, do not fully reflect the genotypes and phenotypes of the respective primary tumors [9]. Fortunately, recent development in this field has led to the implementation of several newer procedures ensuring suitable and gentle dissociation of patient-derived GBM cells, which could preserve genomic profile of the original tissue. Together with their in vitro propagation, either by adherent monolayer culture or by tumor sphere culture in suspension [10], thus obtained cultures have a better potential for testing of anticancer agents tailored to the molecular characteristics of each tumor [9].

The present study was focused on comparison of two methods for GBM cell isolation—mechanic and enzymatic, used in tumor samples of five patients who underwent surgery for GBM. Furthermore, we characterized basic properties of thus derived cell cultures, such as their morphology, proliferation, migration, and expression of selected biological markers. Finally, sensitivity to TMZ and expression of resistance markers in these cells were evaluated and compared. In addition, data obtained from GBM derived cells were contrasted with a well described GBM model—a stabilized A-172 cell line. 

## 2. Results

Five patient-derived GBM samples (GBM1, GBM2, GBM32, GBM33, and GBM34) were obtained and each of them were divided and a small part of the sample was used for cell isolation. The first part was standardly formalin-fixed and paraffin-embedded (FFPE) and the presence of neural markers, such as GFAP and IDH, was immunohistochemically detected. As shown in Figure 1, GFAP and IDH were highly expressed not only in GBM mass, but also in individual cells invading the surrounding tissue. Moreover, GBM cells also had a high positivity for an established proliferation marker ki67. 

The second part of the five GBM samples was used for GBM cell derivation. The samples were again divided into two pieces—one for enzymatic (ED) and one for mechanic dissociation (MD). In thus derived cells, their morphology as well as architecture was examined and compared by phase contrast microscopy. GBM cells derived by ED displayed significant changes in their morphology, including newly displayed cell polarization as well as apparent lamellipodia, suggesting the acquired mesenchymal phenotype (Figure 2A). Conversely, the MD preserved morphology of the neural type in obtained GBM cells, as visible in their elongated cell body and a stellate character (Figure 2B). Also, in thus examined cells the microtubular network reflected similar differences; i.e., a uniform distribution of microtubules in the cytoplasm of the cells originating from MD versus their prominent presence in the cellular protrusions upon ED. Contrary to both types of derived primary GBM cells, the size, morphology, pattern of growth, and spreading, as well as intracellular organization and distribution of microtubules of A-172 cells, significantly differed (Figure 2C).

In all derived GBM cells (ED and MD), the proliferation and migration were next tested with various assays. After 99 h of cultivation, real time analysis by x-CELLigence system, as well as WST-1 proliferation assay, indicated higher proliferation potential in ED GBM cells in comparison with the MD GBM cells (Table 1, Figure 3). Conversely, direct cell counting of selected GBM samples (GBM2 and GBM32) could not prove it (data not shown). The proliferation rate of A-172 cells was comparable to ED GBM cells, however, the shape of the curve resembles more the one of the MD GBM cells. Similarly, unlike MD GBM cells, ED GBM cells showed higher migratory potential after 24 h. Also, the A-172 cell line achieved the highest cell index (proliferation and migration) of all tested GBM cells. 

Higher cell proliferation and migration potentials, as well as different morphology of the cells obtained by ED, were next matched to the analysis of mesenchymal markers N-cadherin and vimentin. The higher abundancy of both proteins was found in ED GBM cells as compared to MD GBM cells, especially in GBM2 sample (Figure 4A). In addition, in ED GBM cells, significantly lower p53 and p21 protein abundances were found in both GBM1 and GBM2 samples. Moreover, lower protein expression of cyclin D1 was detected in MD GBM cells as compared to ED GBM cells, although it was significant in the GBM1 sample only (Figure 4B). In A-172 cells, both mesenchymal markers were expressed to the degree comparable with ED GBM1 cells, while the levels of other analyzed proteins corresponded to the ones detected in MD GBM cells. 

In the following part of the study, the sensitivity of GBM cells to chemotherapeutic drug TMZ was evaluated using WST-1 proliferation assay during 48 h, with subsequent determination of the IC50 value. The MD GBM cells were almost 2.5 times (GBM1), 2.7 times (GBM2), 27 times (GBM33), and 1.5 times (GBM34) less sensitive to TMZ treatment compared to ED GBM cells (Table 2). The only exception concerned the GBM32 sample where lesser sensitivity to TMZ treatment occurred in ED GBM cells. The most sensitive to TMZ treatment in the time interval of 48 h were A-172 cells, whose IC50 value (22 µM) was almost three times lower than one of the most sensitive ED GBM samples (GBM2) (Table 2). Based on these acquired differences, we tested the expression of selected drug resistance markers miR-21, miR-125b, MRP1, and MGMT in the studied cells. Markedly higher expression of both miRNAs was observed in MD GBM1 and GBM2 cells (Figure 5). On the other hand, the detected cellular MRP1 and MGMT protein expression correlated with expression profiles of miR-21 and miR-125b in the GBM1 sample only. In A-172 cells, the level of both tested miRNAs was significantly elevated compared to ED GBM cells, while being more compatible with MD GBM cells. MRP1 and MGMT protein expressions did not significantly differ either. 

Intracellular accumulation of TMZ and its intracellular metabolic conversion to an inactive metabolite AIC was next studied using LC-MS-analysis. As shown in Figure 6, in MD GBM cells a significantly higher amount of TMZ was detected compared to ED GBM cells during 30 min of treatment. The level of AIC was unchanged in all tested time intervals. Accumulation of TMZ inside the A-172 cells was comparable to MD GBM cells, although an intracellular decrease of this parent drug occurred only transiently (in the middle of the followed time interval).

## 3. Discussion

GBM is one of the most aggressive and the most frequent of WHO grade IV primary glial brain cancers in adults. Since surgery has only limited effect due to the infiltrative nature of the tumor, adjuvant concomitant chemotherapy and radiotherapy is implemented in the treatment protocols. Unfortunately, GBM also belongs among the most resistant cancers to available therapies of CNS tumors; i.e., surgery, radiation, and cytotoxic chemotherapy [11]. In order to better understand the biological features of GBM cells and to develop new treatment modalities, cell cultures, whether classically established cell lines or primary cell lines, are used. However, several available procedures for GBM culturing, including an optional use of a range of cultivation media and supplements, are currently used. It is thus clear that the differences in the culturing protocols could affect all significant biological features of the cultivated GBM cells, such as their growth and proliferation, migration, patterns of gene expression, as well as drug resistance [12], thereby resulting in potentially disparate and misleading conclusions. 

In our attempt to find the most effective, easy, and replicable method for primary GBM cancer cell isolation we focused on several published procedures and protocols [9,12]. Gradually, we selected and modified two methods for 2D in vitro derivation of these cells from primary tumor samples; i.e., ED and MD. In the present study, both methods were employed in a set of human GBM tumors and the obtained cells were further tested for their typical GBM-specific phenotypic and genotypic features, such as growth and proliferation, migration, and the expression of several markers important for drug sensitivity/resistance. In addition, as a background control, the known stabilized GBM cell line A-172 was used.

Our results showed that both ED and MD approaches enable successful establishment of primary GBM cell lines, which could be maintained under standard laboratory conditions and subjected to various tests and assays. On the other hand, significant differences in ED- and MD-dependent GBM cell phenotypes and behaviors were detected, which include cell appearance, growth, migration, as well as the expression of select cell cycle-specific molecules. In all analyzed primary GBM cells, the ED method always yielded larger cells with higher migration activity as compared to MD approach. Moreover, these differing biological activities also corresponded to an increased presence of typical mesenchymal markers vimentin and N-cadherin in ED-derived cells, thereby suggesting the putative occurrence of epithelial-to-mesenchymal transition (EMT) in cultured cells. More mesenchymal-like phenotypes of ED-derived GBM cells compared to MD-derived cells were also obvious from their morphology and internal organization of microtubular networks. Our other obtained results also show that A-172 cells are highly dissimilar in their morphology to the primary GBM cells; i.e., they are smaller and lack mesenchymal markers, despite the fact that their proliferation and migration are considerable. Taken together, these results suggest that although both ED and MD approaches may be used for establishment of primary GBM cell lines, they generate GBM cell populations differing significantly in their phenotype and biological behavior. 

Standard treatment of newly diagnosed GBM consists of surgery, followed by adjuvant chemotherapy and radiotherapy regimens. However, even in gross total surgical removal of the tumor mass with the most aggressive adjuvant treatment protocols, the median patient survival time is only 14.6 months [13]. Therapy failures of this malignant disease likely depend on its inherent complexity and numerous mechanisms of drug resistance [14]. One of them is the existent mesenchymal phenotype of malignant cells, previously described in different types of cancer, for example in breast cancer [15], lung cancer [16], or in pancreatic cancer [17]. Still, in our study, the cells with mesenchymal phenotypes, obtained via ED, proved to be much more sensitive to TMZ than non-mesenchymal MD cells. This paradoxical observation seems to suggest that EMT itself, or a degree of it present in the analyzed cells, is not the only determinant of TMZ sensitivity of primary GBM cells, as suggested by our observed sensitivity to TMZ of A-172 cells when compared to all tested primary GBM cells. Another possible explanation of this discrepancy may relate to the rate of cell proliferation. TMZ is known to damage DNA in exposed cells, leading to cell cycle arrest at G2/M phase, and finally to apoptosis [18]. Slower-growing MD primary GBM cells would therefore be less responsive to this cytotoxic treatment than faster proliferating ED primary GBM cells due to a differing dependence of cells on DNA replication. Conversely, the activity of O6-methylguanine DNA methyl transferase MGMT, which helps in repairing TMZ-induced DNA damage and whose expression acts as a predictor of TMZ efficacy in GBM, was not clearly found to play any significant role in our model. 

The efflux of cytotoxic drugs through the cellular membrane can cause multiple resistances by multi-drug resistance proteins, such as ATP-binding cassette (ABC) transporter family, which includes *P*-glycoprotein (*P*-gp), multi-drug resistance protein (MRP), and breast cancer resistance protein (BCRP). [14]. In our study, the expression of MRP1 in GBM cells did not match the drug sensitivity profiles and thus we cannot safely conclude whether this, or other resistance proteins, significantly contributed to the final drug response in the present model. Still, TMZ is also rapidly metabolized both extra- and intracellularly via irreversible and pH-dependent degradation to MTIC and then to AIC [19]. In this respect, in TMZ resistant GBM cells (MD derived) a higher level of AIC was detected when compared to sensitive ED GBM cells, which indicates the resistance to TMZ could also eventually be caused by faster intracellular degradation of TMZ. In addition, this assumption is also corroborated by similar observations on A-172 cell line. 

Recently, several miRNAs (let 7, miR-9, miR-17, miR-21, miR-31, or miR-125b) have been described to be important regulators of drug resistance in GBM [20]. To this end, miR-21 was already recognized as potential biomarker of acquired TMZ resistance in GBM cell line D54MG, when miR-21 inhibitor treatment, together with TMZ exposure, significantly increased the rate of apoptosis compared to TMZ treatment alone [21]. The inhibitory effect of overexpressed miR-21 on TMZ-induced apoptosis was described in other GBM cell line, U87MG, where it was mediated through downregulation of proapoptotic proteins, Bax and caspase-3, as well as via upregulation of antiapoptotic protein Bcl-2 [22]. Accordingly, the higher expression of miR-21, as well as miR-125b, was detected in primary MD GBM cells, where it correlated with decreased sensitivity to TMZ. Similar correlation, however, was not found in case of A-172 cells, which were, in this instance, different compared to examined primary MD GBM cells. 

## 4. Material and Methods

### 4.1. Media

Transport medium for tumor samples: PBS 10× with calcium and magnesium, aqua pro injectione. Washing medium: PBS 10× w/o calcium and magnesium, aqua pro injectione. Cultivation medium: RPMI 1640 medium (Sigma-Aldrich, Prague, Czech Republic), 15% fetal bovine serum (Gibco, Thermo Fisher Scientific, Waltham, MA USA), insulin (100 IU/mL, Eli Lilly, Czech Republic), transferrin (2 mg/mL, Sigma-Aldrich).

### 4.2. Clinical Samples

All clinical samples were obtained from patients who underwent surgery for GBM at University Hospital in Hradec Králové. The study was approved by Ethics Committee, University Hospital in Hradec Kralove (Reference No. 201709 S13P—attached as a Appendix A) and patients gave their written consent. Subjects with different clinical stage and grade of the disease were included into the study. The amount and quality of material sampled for cell line derivation varied based on the features of the surgical specimen. GBM samples from 5 patients were chosen (samples GBM1, GBM2, GBM32, GBM33, GBM34) for cultivation and analysis of resulting cell lines. Cell isolation protocols from tumor samples and further cell handling is described below. 

### 4.3. Mechanic Dissociation

Samples of a tumor tissue were cut into small pieces and then transported in a tube with a transport medium at room temperature. The tissue was washed in a Petri dish several times to remove erythrocytes. The sample was next cleansed from necrotic tissue and mechanically disintegrated by scissors and a scalpel. Pulverized tissue was then pressed through the syringe. Cells were collected with a Pasteur pipette, resuspended in cultivation medium, and planted into the cultivation flasks.

### 4.4. Enzymatic Dissociation

The tissue in transport medium was placed on a Petri dish and washed several times to remove erythrocytes. The sample in the Petri dish was cleansed from fat and necrotic tissue and mechanically disintegrated by scissors and a scalpel for smaller pieces. Slices of tissue were treated with Accumax™ solution (Sigma Aldrich) for 5 to 20 min, based on the size of the tissue, at room temperature. The disintegrated tissues were further homogenized using the syringe, transported into the centrifugation tube, and centrifuged at 300× *g* for 5 min. The supernatant was removed and pellet was resuspended in cultivation medium and then transferred into the cultivation flasks.

### 4.5. Cultivation of the Cells 

Cells were grown in cultivation flasks in a humidified atmosphere containing 5% CO_2_ at 37 °C. Cells were usually checked 2 days after isolation and fresh medium was added every 3 days for two weeks until the cells formed monolayer. Passaging of the cells was carried out with trypsin/EDTA solution at 37 °C for 5–10 min. Thereafter, cells were washed using fresh cultivation medium (RPMI/15% FBS) and seeded into the cultivation flasks or microtiter plates. All experiments in this study were performed in passages 2, 3, and 4. We did not observe any significant differences and epithelial-to-mesenchymal transition (EMT)-like mechanisms were not detected in these low passages.

### 4.6. Freezing and Thawing 

Cell suspension was centrifuged for 5 min at 500× *g* and cell pellet was mixed with 10% DMSO in FBS to a final concentration of 2 × 10^6^ cells/mL. The cell culture was stored in liquid nitrogen (−180 °C) for further analysis. Thawing of the samples was rapidly done in water bath at 37 °C, cells were mixed with cold cultivation medium, centrifuged for 5 min at 500× *g*, and then the cell pellet was transferred in a cultivation flask with fresh medium. Further analysis was carried out in cells if their viability reached at least 80%.

### 4.7. Cytotoxicity Assay

Viability of GBM primary cells and A-172 cell line after TMZ treatment was evaluated by WST-1 test. This test is a colorimetric assay based on the cleavage of the tetrazolium salt to colored formazan by mitochondrial dehydrogenases in viable cells. The test thus quantifies cell proliferation and viability based on the rate of mitochondrial enzymes activity. GBM primary cells and A-172 cells were plated in 96-well microtiter plates for 24 h and then exposed to TMZ at various concentrations in RPMI cultivation medium for up to 48 h. At the end of 48 h time interval, 100 μL of WST-1 solution (1:20 final dilution) was added and the absorbance was recorded after 2 h incubation at 450 nm with 650 nm of reference wavelength by spectrophotometer Tecan Infinite M200 (Tecan, Switzerland). 

### 4.8. Cell Proliferation Assay

Cell proliferation of GBM primary cells and A-172 cell line was performed using xCELLigence Real-Time Cell Analyzer. Microelectrodes integrated in the bottom of E-plates measure electrical impedance, which monitors cell proliferation and viability in real time without any labeling. The relative change is displayed by the system as arbitrary units called “cell index”. First, plates with 90 µl of medium in each well were inserted in the device for background measurement. Then, cell suspension (2000 cells/well in 100 µL of medium) was added in quadruplicate to the appropriate wells. The attachment, spreading, and proliferation of the cells were monitored every 1 h for 99 h of cultivation. 

### 4.9. Cell Migration Assay

Analyses of cell migration of primary GBM cells and A-172 cell line were carried out in CIM-plate 16 using xCELLigence Real-Time Cell Analyzer. CIM-plate 16 is a modified Boyden chamber composed of an upper chamber and a lower chamber. Upper chambers of CIM-plates 16 were filled with serum-free medium supplemented with 1% BSA and into the lower chambers medium supplemented with 10% FBS was added. Then, the upper and lower chambers were locked together to form a tight seal. The plate was pre-incubated for 1 h in the CO_2_ incubator at 37 °C before obtaining a background measurement. Optimal number of the cells (30,000/100 µL), resuspended in serum-free medium supplemented with 1% BSA, was added to each well of the upper chamber and cell index values were recorded every 10 min for 24 h.

### 4.10. RNA Extraction and cDNA Synthesis

First, grown GBM primary cells and A-172 cells were collected from 6-well plates at a density of 150,000 cells/mL using TriReagent (Sigma Aldrich, Czech Republic). Total RNA was isolated according to manufacturer’s instructions using Direct-zol RNA MiniPrep kit (ZymoResearch, Irvine, CA, USA). RNA yields and purity were measured using NanoDrop ND-2000 UV-Vis Spectrophotometer (Thermo Scientific, Czech Republic). Absorption ratio A260/A280 of all tested samples was greater than 1.8. The quality of RNA was checked by Agilent 2100 Bioanalyzer and the RNA integrity number (RIN) was greater than 8. For the cDNA synthesis of miRs, the reaction mixture included a mix of stem-loop oligos specific for each miR tested (miR-21, miR-125b, miR-16). First, strand synthesis was carried out using ProtoScript II reverse transcriptase according to the manufacturer’s instruction (New England Biolabs). After initial heat denaturation of 0.5 µg of total RNA (65 °C for 5 min), the reactions (10 µL) were incubated for 30 min at 16 °C, for 30 min at 42 °C, and for 5 min at 80 °C. Obtained cDNA was diluted (50 x) prior to qPCR. All cDNAs were stored at −20 °C until qPCR assay.

### 4.11. Primer Design, Quantitative Real-Time RT-PCR 

The primers for miRNA quantification were designed manually and checked using OligoCalc (http://www.basic.northwestern.edu/biotools/oligocalc.html). The primers for miRNA analysis were synthesized by Generi Biotech, Czech Republic and are available upon request. The qPCR analyses were carried out in Lightcycler96 Real-Time PCR Detection System (Roche, Berlin, Germany), using SYBR Green I detection. The reaction mixture was prepared according to manufacturer’s instructions (Fast Start Universal SYBR Green Master (Rox)—Roche Life Science, Berlin, Germany). Both forward and reverse primers were, in final concentrations, 100 nM. The PCR reactions were performed as described in [23]. 

All qPCRs runs were performed in duplicates. Calculations were based on the “Delta–Delta Ct method” with the data expressed as fold change of the cell cultures relative to the control. Beta 2-microglobulin (B2M) miR-16 was used as a reference gene for miR analysis.

### 4.12. Western Blot Analysis

Confluent primary GBM and A-172 cells were washed with ice-cold PBS and collected in ice-cold lysis buffer (RIPA buffer). The lysates were resuspended and the quantity of total protein in supernatant was determined by BCA assay. Samples were boiled in SDS sample buffer (Tris–HCl pH 6.8, 40% glycerol, 6% SDS, 0.2 M DTT, 0.1 g bromphenol blue) for 5 min/95 °C and in concentration 30 µg of total protein per well were loaded onto SDS/polyacrylamide gel. Separated proteins were transferred to a PVDF membrane (100 V, 90 min). Then, membranes were incubated with TBST milk solution for 1.5 h at room temperature. Primary antibodies were diluted according to the manufacturer’s protocols in TBST containing 2% of milk or 2% of BSA (polyclonal rabbit anti-*N* cadherin, 1:1500; polyclonal rabbit anti-vimentin, 1:2000; monoclonal mouse anti-p53 antibody, 1: 1500; monoclonal rabbit anti-MRP1/ABCC1, 1:2000; monoclonal rabbit anti-cyclin D1, 1: 2000 – Cell signaling technology; polyclonal rabbit anti-p21, 1:2500—Santa Cruz Biotechnology; monoclonal mouse anti-β actin, 1:10,000-Sigma Aldrich). After incubation with primary antibodies at 4 °C overnight, membranes were washed in TBST. Then, the membranes were incubated with secondary peroxidase-conjugated antibodies (1:20,000, 2 h, 25 °C), followed by six washes with TBST buffer. The detection of signal was developed using a chemiluminescence ECL Prime Western Blotting Detection Reagent (Amersham, GE Healthcare Life Science, Little Chalfont, Great Britain). Relative quantification of chemiluminescence was evaluated using Imaging System (Gel Logic 2200 Pro, Bruker BioSpin, Ettlingen, Germany).

### 4.13. Immunohistochemical Analysis

First, obtained surgical samples were fixed in formalin and paraffin blocks were prepared, using standard procedure. Sections (2–3 µm thick) were cut from these blocks and mounted on positively charged slides. The primary antibodies used for assay were GFAP and Ki67 (Ventana, ready to use). Antigen retrieval was performed on board (Ventana Benchmark Ultra machine), using high pH and detection by Ventana Ultraview (for GFAP) and Venta Optiview (for Ki67) detection kit. IDH1 was performed by Dianova, 1:50 and antigen retrieval was performed in PT link (Dako), high pH with detection by FLEX Dako in Dako autostainer. Finally, slides were counterstained with haematoxylin.

### 4.14. MS Analysis

Isolated GBM-MD, GBM-ED, and A-172 cells were plated in 6-well plates in density 150,000 cells/mL and then incubated in RPMI-1640 medium containing 500 µM concentration of TMZ for 4 h. The concentration of DMSO in medium was 0.1%. The cells were collected into the 500 µL of sterile distilled water. The amount of total proteins in the homogenates from the cell suspension was measured using a bicinchoninic acid (BCA) assay (Sigma Aldrich) according to the manufacturer’s protocol.

The cell suspension or medium, respectively, were mixed with acetonitrile and methanol in ratio 1:1:1, vortexed for 15 min, and centrifuged at 13,000 rpm for 4 min. The supernatant was filtered through the 0.22 µm PTFE syringe filter to the glass vial ready for analysis. One microliter of the sample was used for the subsequent UHPLC-MS/MS analysis.

Detection of TMZ and its metabolites 5-(3-methyltriazen-1-yl)imidazole-4-carboxamide (MTIC) and 5-aminoimidazole-4-carboxamide (AIC) was performed on the Agilent 1290 Infinity II UHPLC system coupled to the Agilent 6470 QqQ mass spectrometer. Chromatographic conditions were maintained at gradient elution of 0.4 mL/min by 0.1% formic acid in water and methanol (0–0.5 95:5, 0.5–3.0 gradient to 5:95, 3.0–4.0 5:95, 4.0–5.0 95:5), thermostated autosampler set to 15 °C, and column thermostat equipped with the Zorbax Eclipse plus RRHD C18 2.1 × 50 mm, 1.8 µm (PN 959757-902) column kept to 30 °C. MS source parameters were set to the following: drying gas 200 °C at 2 L/min, sheath gas 400 °C at 12 L/min, nebulizer pressure 25 psi, capillary voltage 2500 V. Transitions of [M+H]+ ions m/z were detected with setting of dwell time 50 ms, cell acceleration 4 V, and fragmentor 88 V for TMZ 195→138 and 55 (collision energy-CE 8 and 28 V), MTIC 169→109 and 43 (CE 20, 40 V), and AIC 127→110, 82 and 55 (CE 20, 20, 40 V).

### 4.15. Phase Contrast Microscopy

Morphology and architecture of primary GBM and A-172 cells was recorded in standard tissue culture flasks under Olympus IX-70 phase contrast microscope using various magnifications.

### 4.16. Immunofluorescence and Image Analysis

Primary GBM and A-172 cells were grown in cytospine chambers for 48 h, then fixed with 2% paraformaldehyde (20 min, 25 °C), rinsed with PBS permeabilized, and blocked with 1% Triton X and 5% BSA in PBS (30 min, room temperature). The cells were incubated with a primary antibody against α-tubulin (1:100, polyclonal rabbit—Cell signaling technology) at 4 °C overnight. Then, the cells were washed three times with cold PBS (5 min, 25 °C) and were incubated for additional 1 h (room temperature) with Alexa Fluor 488-labelled goat anti-mouse antibody. Thereafter, the cells were rinsed three times with PBS and labelled with DAPI (10 µg/mL). The specimens were mounted into the Prolong Gold mounting medium (Invitrogen-Molecular Probes, Inc., Carlsbad, California, CA, USA) and examined using fluorescence microscopy technique (Nikon Eclipse E 400 (Nikon Corporation, Kanagawa, Japan)). The results were analyzed using LUCIA DI Image Analysis System LIM 4.2 (Laboratory Imaging Ltd., Prague, Czech Republic). All the samples were tested in duplicates in three independent experiments.

### 4.17. Statistical Analysis

Data are presented as the mean ± SD of averages from at least two experiments. Statistical analysis was performed using GraphPad Prism 6.0 Software using two-way analysis of variance (ANOVA), followed by Sidak’s multiple comparison test. Statistical significance was acceptable to a level of *p* ˂ 0.05. The concentration inducing 50% decrease of cell viability (IC50) was calculated using nonlinear regression by GraphPad Prism 6.0.

### 4.18. Ethical Approval

All procedures performed in studies involving human participants were in accordance with the ethical standards of the institutional and/or national research committee (University Hospital Hradec Kralove Ethics Committee—reference number 201709 S13P) and with the 1964 Helsinki declaration and its later amendments or comparable ethical standards. Informed consent was obtained from all individual participants included in the study.

## 5. Conclusions

In our work we showed that primary GBM cells from patient tumor samples could be obtained both by ED and MD approaches, although biological behavior and properties of these cells significantly differ. Unlike in primary MD GBM cells, higher proliferation, as well migration, was observed in primary ED GBM cells, which were also associated with the acquired mesenchymal phenotype and higher sensitivity to TMZ. Observed differing sensitivity to TMZ in both types of cells was further associated with the varying expression of MRP1 efflux transporter, particular microRNAs, as well as intracellular metabolism of TMZ. Finally, the same analyses of stabilized GBM cell line A-172, which is an established model in GBM research, revealed some similarities. On the other hand, several important differences in measured parameters in this cell’s line, with the most important one being sensitivity to TMZ, were noted. Thus, in case of studies into biology of EMT in GBM on samples derived from patients, an MD approach to their isolation should be the preferred method of primary GBM cell treatment, due to better preservation of original biological features of tumor cells. Still, care has to be taken when interpreting data from cell line A-172 in these type of studies, since they show considerable heterogeneity and sometimes contrasting phenotypes and behavioral patterns, when compared to primary GBM cells. 

## Figures and Tables

**Figure 1 ijms-20-04630-f001:**
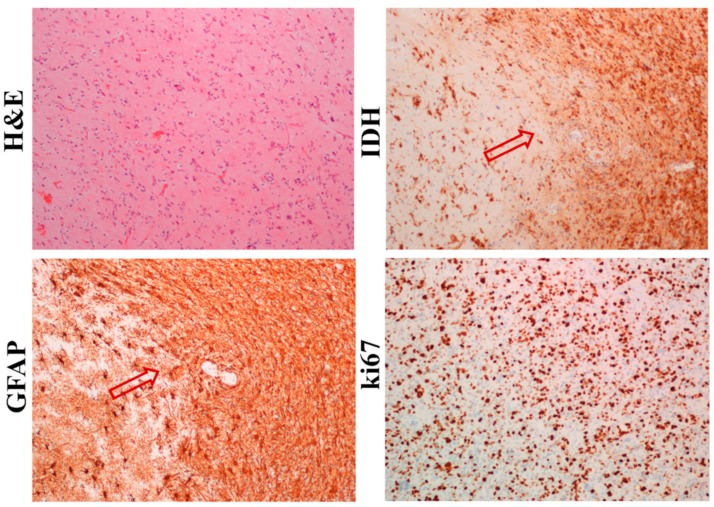
Immunohistochemical analysis of the glioblastoma (GBM) sample (sample No. 1). GBM samples were processed and GFAP, IDH, ki-67, and haematoxylin and eosin were detected by immunohistochemistry and histochemistry, as described in the Materials and Methods section. Bright-field microscopy, 100×, scale. Arrows show the GBM margin.

**Figure 2 ijms-20-04630-f002:**
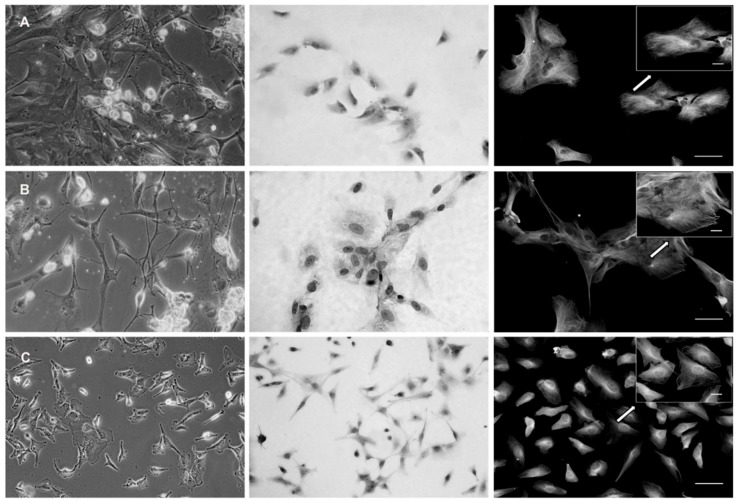
Phase contrast (**left**), haematoxylin and eosin (**middle**), and fluorescence (microtubules) (**right**) of primary glioblastoma cells (GBM1) isolated using enzymatic method (**A**) and mechanic method (**B**). Stabilized glioblastoma cell line A-172 (**C**) was used for comparison. Magnification 200 ×, scale 30 µm, inset images—magnification 400×, scale 5 µm.

**Figure 3 ijms-20-04630-f003:**
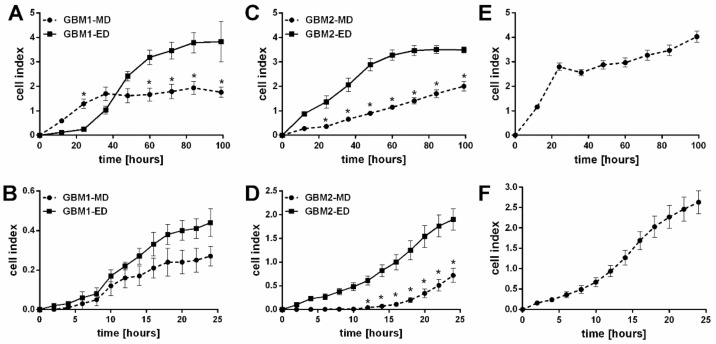
Proliferation (**A**,**C**,**E**) and migration potential (**B**,**D**,**F**) of primary glioblastoma cells (GBM) cells obtained from five patient samples (GBM1, GBM2, GBM32, GBM33, GBM34) with enzymatic method (ED) or mechanic method (MD) and stabilized A-172 cell line. Proliferation assay (recorded every h for 99 h) and migration assay (recorded every 10 min for 24 h) were performed using xCELLigence Real-Time Cell Analyzer as described in the Materials and Methods section. Each measurement was performed in three independent experiments. * *p* < 0.05 ED vs. MD; data are expressed as average ± SD.

**Figure 4 ijms-20-04630-f004:**
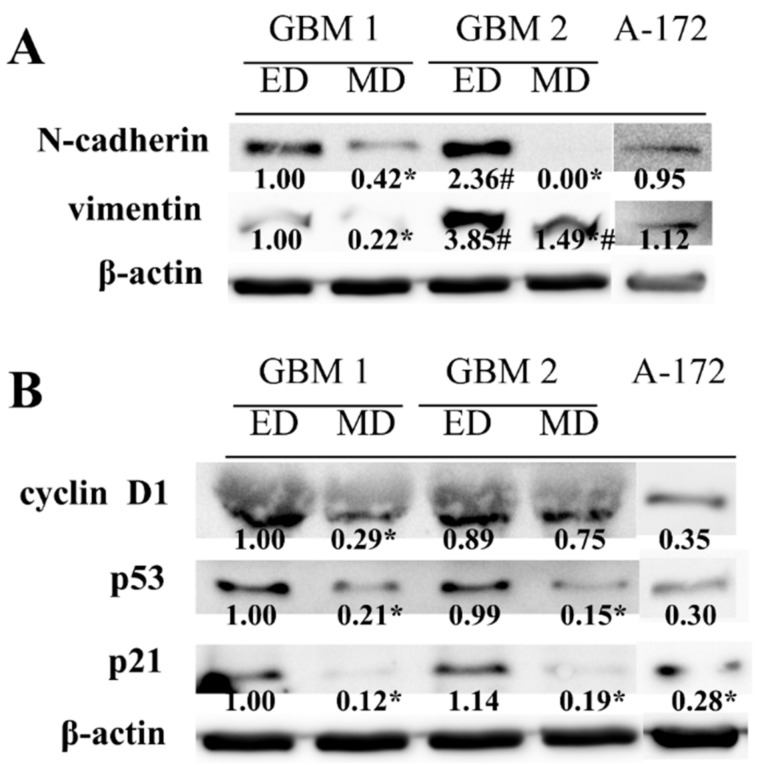
The comparison of protein expression in primary glioblastoma (GBM) cells obtained using enzymatic method (ED) and mechanic method (MD) from the GBM of two patients and in stabilized cell line A-172. Expression of mesenchymal markers, N-cadherin and vimentin (**A**), as well as cell cycle relevant cyclin D1, p53, and p21 (**B**) was determined by Western blot analysis. β-actin was used as a loading control. Measurements were performed in three independent experiments. * *p* < 0.05 ED vs. MD; # *p* < 0.05 ED vs. MD in GBM1 vs. GBM2.

**Figure 5 ijms-20-04630-f005:**
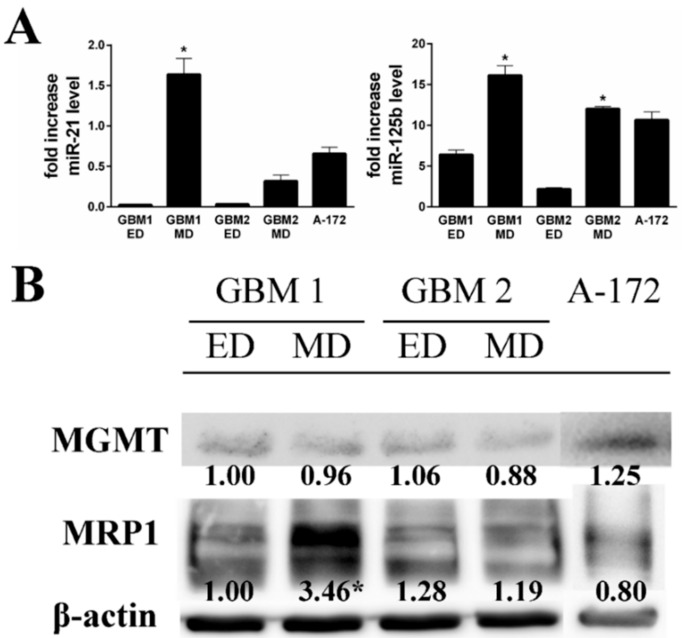
The expression of selected drug resistance markers in primary glioblastoma cells, GBM1 and GBM2, isolated using enzymatic method (ED) and mechanic method (MD) from the GBM of two patients and in stabilized cell line A-172. The expression of miR-21 and miR-125b was determined by RT-PCR (**A**). Data are expressed as fold increase ± SD of averages from two independent experiments. Expression of MRP1 and MGMT proteins were determined by Western blot analysis (**B**). β-actin was used as a loading control. Measurements were performed in three independent experiments. * *p* < 0.05 ED vs MD.

**Figure 6 ijms-20-04630-f006:**
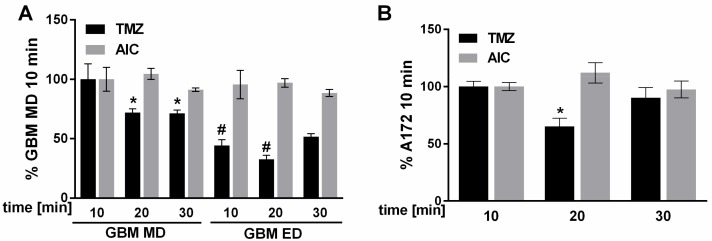
MS analysis of parent drug TMZ and its 5-aminoimidazole-4-carboxamide (AIC) metabolite in primary glioblastoma cells, isolated using enzymatic method (ED) and mechanic method (MD) from the GBM **(A)** and in stabilized cell line A-172 **(B)**, used for comparison. The data are expressed as percentage of TMZ or AIC inside the GBM-MD, GBM-ED, and A-172 cells per mg of protein and amount was determined by LC-MS analysis after 10, 20, and 30 min treatment. Measurements were performed in two independent experiments. ** p* < 0.05 20 and 30 min intervals vs 10 min interval; *# p* < 0.05 ED vs. MD.

**Table 1 ijms-20-04630-t001:** The cell index value and the cell viability of five primary glioblastoma cells (GBM1, GBM2, GBM32, GBM33, and GBM34) isolated using enzymatic method (ED) and mechanic method (MD) from the GBM of five patients during 99 h (cell proliferation) and during 24 h (cell migration). Results were compared with stabilized cell line A172. * Absorbance (cell viability) was determined after 72 h cultivation using WST-1 analysis. *p* < 0.05 enzymatic vs. mechanic dissociation; data are expressed as average ± SD.

Cell Line	Cell Index		Cell Viability
	Proliferation	Migration	Absorbance
**GBM1-ED**	3.8 ± 0.8	0.4 ± 0.07	0.89 ± 0.12
**GBM1-MD**	1.8 ± 0.2 *	0.3 ± 0.05	0.80 ± 0.10
**GBM2-ED**	3.5 ± 0.1	1.9 ± 0.2	0.92 ± 0.11
**GBM2-MD**	2.0 ± 0.2 *	0.7 ± 0.2 *	0.63 ± 0.06 *
**GBM32-ED**	4.3 ± 0.3	2.3 ± 0.1	1.41 ± 0.13
**GBM32-MD**	3.9 ± 0.3	1.3 ±0.2 *	1.19 ± 0.09 *
**GBM33-ED**	3.3 ± 0.3	2.3 ± 0.2	0.90 ± 0.08
**GBM33-MD**	2.8 ± 0,3	1.4 ± 0.2 *	0.73 ± 0.08
**GBM34-ED**	3.0 ± 0.2	0.6 ± 0.1	1.32 ± 0.10
**GBM34-MD**	2.2 ± 0.4	0.4 ± 0.1	1.28 ± 0.12
**A172**	4.0 ± 0.4	2.6 ± 0.3	1.42 ± 011

**Table 2 ijms-20-04630-t002:** The concentration inducing 50% decrease of cell viability (IC50) values of temozolomide (TMZ) treatment in primary glioblastoma cells (GBM1, GBM2, GBM32, GBM33, and GBM34), obtained using enzymatic dissociation (ED) and mechanic dissociation (MD) from appropriate human samples after 48 h of incubation. Results were compared with stabilized cell line A172. * *p* < 0.05 enzymatic vs. mechanic dissociation; data are expressed as average ± SD.

Cell Line	IC50 [µM]
**GBM1-MD**	362.20
**GBM1-ED**	139.50 *
**GBM2-MD**	174.60
**GBM2-ED**	63.46 *
**GBM32-MD**	1241.00
**GBM32-ED**	2476.00 *
**GBM33-MD**	2644.00
**GBM33-ED**	97.60 *
**GBM34-MD**	767.10
**GBM34-ED**	518.80
**A172**	22.28

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
