# Peer review of "The Evaluation of Glioblastoma Cell Dissociation and Its Influence on Its Behavior"

_ijms, 2019, doi:10.3390/ijms20184630_

Round 1

Reviewer 1 Report

In this manuscript, the authors examined whether two different methods of glioblastoma cell isolation affect cell proliferation, migration, gene expression, and drug resistance in in vitro culture. The aim of this study is very important, but most data are obtained from two GBM samples, and it is not easy to understand the conclusion of this study that the mechanic method approach should be the preferred method of primary GBM cell isolation. In addition, it is also difficult to know what is several important differences in A172 cells from the results. In particular, it is not unclear whether intracellular organization and distribution of microtubules in A172 cells significantly differ in the present images. Therefore, I do not recommend the publication of this manuscript in its present form.

Author Response

We carefully consider the reviewer comments and revised our manuscript according to these comments. The corrections are highlighted in revised manuscript in yelow or using the tool "Revisions".

The aim of this study is very important, but most data are obtained from two GBM samples, and it is not easy to understand the conclusion of this study that the mechanic method approach should be the preferred method of primary GBM cell isolation.

The conclusion section is complemented by explanation of advantage of mechanical dissociation, as highlighted in text.

In addition, it is also difficult to know what is several important differences in A172 cells from the results.

We have added the requested information on differences between A-172 cells and primary GBM cells to the body text. These differences are also highlighted in yellow throughout the text.

In particular, it is not unclear whether intracellular organization and distribution of microtubules in A172 cells significantly differ in the present images.

Figure 2 is complemented by higher magnification of microtubules in A-172 and primary GBM cells as well.

Reviewer 2 Report

This manuscript entitled “The evaluation of glioblastoma cell dissociation and its influence on its behavior.” shows that primary glioblastoma cells isolated with mechanical dissociation demonstrate higher proliferative and migrative capabilities and stronger chemoresistance to temozolomide than those isolated with enzymatic dissociation. This article is considered of value in that importance of tissue dissociation method in primary culture is recognized, and dissociation without enzymatic treatment could be better approach to obtain cell lines which reflect true phenotype of primary tumors. However, this reviewer has some questions before recommendation for publication. These are listed below:

(1) Figure 5AB

  Sensitivities to TMZ are largely dependent on methylation status of MGMT gene in most GBMs. If elimination of drug by MRP1 and miR-21 signals are more important than transferase activity of MGMT, epigenetic status of MGMT and results of both MGMT-methylated and unmethylated cells should be indicated.

(2) Figure 3AC

  Proliferative velocity and saturation density are different factors in proliferation. Saturation density appears to be decreased in GBM1-ED in Fig A. How was the size and stratification of the cells changed? Not only indirect method but also actual cell counting is required.

(3) Figure 1

  Photos with higher magnification are (also) required.

(4) Figure 2

  It should be indicated how many passages the cells in the photos 2A and 2B are cultured through. Did the shapes and size of the cells changed through passages associated with EMT-like mechanism?

Author Response

We carefully consider the reviewer comments and revised our manuscript according to these comments. The corrections are highlighted in revised manuscript using the tool "Revisions".

(1) Figure 5AB

Sensitivities to TMZ are largely dependent on methylation status of MGMT gene in most GBMs. If elimination of drug by MRP1 and miR-21 signals are more important than transferase activity of MGMT, epigenetic status of MGMT and results of both MGMT-methylated and unmethylated cells should be indicated.

We are aware that epigenetic status of MGMT would improve our study, but at this moment we have no established methodology for methylated/unmethylated status determination.

(2) Figure 3AC

Proliferative velocity and saturation density are different factors in proliferation. Saturation density appears to be decreased in GBM1-ED in Fig A. How was the size and stratification of the cells changed? Not only indirect method but also actual cell counting is required.

The data in Table 1 are complemented by data from WST-1 analysis – absorbance measurement of control (untreated cells) after 72h incubation. The same amount of the cells (MD GBM and ED GBM) were plated for each experiment. However, the growing curve of tested cells was heterogenous.

Cell index

Cell viability

proliferation

migration

absorbance

GBM1 - ED

3.8 ± 0.8

0.4 ± 0.07

0.89 ± 0.12

GBM1 - MD

1.8 ± 0.2*

0.3 ± 0.05

0.89 ± 0.10

GBM2 -ED

3.5 ± 0.1

1.9 ± 0.2

0.92 ± 0.11*

GBM2 -MD

2.0 ± 0.2*

0.7 ± 0.2*

0.63 ± 0.06

GBM32 -ED

4.3 ± 0.3

2.3 ± 0.1

1.41 ± 0.13

GBM32 -MD

3.9 ± 0.3

1.3 ±0.2*

1.19 ± 0.09

GBM33 -ED

3.3 ± 0.3

2.3 ± 0.2

0.90 ± 0.08

GBM33 -MD

2.8 ± 0,3

1.4 ± 0.2*

0.73 ± 0.08

GBM34 -ED

3.0 ± 0.2

0.6 ± 0.1

1.32 ± 0.10

GBM34 -MD

2.2 ± 0.4

0.4 ± 0.1

1.28 ± 0.12

A172

4.0 ± 0.4

2.6 ± 0.3

1.42 ± 011

(3) Figure 1

Photos with higher magnification are (also) required.

Higher magnification of images in Figure 1 was used.

(4) Figure 2

It should be indicated how many passages the cells in the photos 2A and 2B are cultured through. Did the shapes and size of the cells changed through passages associated with EMT-like mechanism?

All experiments in this study were performed in passages 2, 3 and 4. We didn’t observed any significant differences through these passages. In lower passages, EMT-like mechanisms were not found out. On the other hand, cells in passages 7 and higher showed several changes. However, cells in higher passages were not used for this study.

Round 2

Reviewer 1 Report

I would like to recommend the publication of the revised manuscript.

Author Response

Reviewer 1 had no further comments or suggestions to our manuscript.

Reviewer 2 Report

This manuscript has been improved, however, this reviewer still have some questions.

(1) Figure 5AB

This reviewer accepts the author’s mention although methylation status of MGMT is the most important factor for sensitivity to TMZ.

(2) Figure 3AC

Proliferation velocity should be evaluated in the logarithmic phase, and that of GBM1-MD is rather increased compared with GBM1-ED although the saturation density of GBM1-MD is significantly decreased in Figure 3A. Increased saturation density suggests morphological change with cellular enlargement or decreased stratification. This reviewer thinks the result of saturation density is compatible with decreased mesenchymal markers shown in Figure 4A, however, actual cell number (actual counting) and careful observation for morphology are desirable at least in GBM1.

(3) Figure 1

This figure has been improved.

(4) Figure 2

Information of the passage should be described in Material and Method.

Author Response

Dear reviewer,

We carefully consider your comments and this revisions and answers are attached below:

(1) Figure 5AB

This reviewer accepts the author’s mention although methylation status of MGMT is the most important factor for sensitivity to TMZ.

We will focus for evaluation of this methodology in the future.

(2) Figure 3AC

Proliferation velocity should be evaluated in the logarithmic phase, and that of GBM1-MD is rather increased compared with GBM1-ED although the saturation density of GBM1-MD is significantly decreased in Figure 3A. Increased saturation density suggests morphological change with cellular enlargement or decreased stratification. This reviewer thinks the result of saturation density is compatible with decreased mesenchymal markers shown in Figure 4A, however, actual cell number (actual counting) and careful observation for morphology are desirable at least in GBM1.

As suggested, we have performed direct counting of GBM-ED and GBM-MD cells. We have selected GBM2 and GBM32 samples due to the original statistically different proliferation rates as measured by WST-1 assay. Unfortunately, we could not do it with GBM1 sample, as these cells were completely depleted by other requested experiments. Still, the results from GBM2 and GBM32 samples show that there was no statistical difference in the cell numbers, while both samples were counted in the log phase. This finding thus confirms the observation of the reviewer about different sizes of both types of the cells (GBM-MD vs GBM-ED). The results section of the manuscript has been upgraded accordingly.

Description of the method and obtained results are attached below:

Cell counting:

GBM primary cells GBM2 and GBM32 (ED GBM and MD GBM variant of each) were plated and cultivated in 6-well plates in density 150.000 cells/well for up to 48 h (time corresponded to log phase of the curves). At the end of this interval, the cells were trypsinised and resuspended in fresh cultivation medium. Cell suspension was mixed with trypan blue solution in PBS (4 mg/ml, Sigma Aldrich). After 2 min incubation at room temperature, viable cells (unstained cells) were counted using Cellometer Auto T4 Automated Cell Counter (Nexcelom Bioscience). Each sample was assayed in three parallels and two independent experiments were performed.

Tab.1: Cell number of two primary glioblastoma cells (GBM2, GBM32) isolated using enzymatic method (ED) and mechanic method (MD) from the GBM of two patients during 48 h.

GBM2

GBM32

ED

MD

ED

MD

[cell number x 104]

7.10 ± 0.34

6.45 ± 0.27

6.73 ± 0.23

6.48 ± 0.19

%

100.00 ± 4.72

90.89 ± 3.75

100.00 ± 3.24

96.29 ± 2.68

(3) Figure 1

This figure has been improved.

(4) Figure 2

Information of the passage should be described in Material and Method.

We have added the requested information into the Material and methods section – Cultivation of the cells.

Round 3

Reviewer 2 Report

This manuscript has been improved. This reviewer would expect future study that clarifies relationship between cellular dissociation methods and EMT phenotypes and their mechanism and which method reflects actual biology of glioblastoma in vivo.